# First ATR-FTIR Characterization of Black, Green and White Teas (*Camellia sinensis*) from European Tea Gardens: A PCA Analysis to Differentiate Leaves from the In-Cup Infusion

**DOI:** 10.3390/foods13010109

**Published:** 2023-12-28

**Authors:** Elisabetta Giorgini, Valentina Notarstefano, Roberta Foligni, Patricia Carloni, Elisabetta Damiani

**Affiliations:** 1Department of Life and Environmental Sciences, Università Politecnica delle Marche, Via Brecce Bianche, I-60131 Ancona, Italy; e.giorgini@univpm.it (E.G.); v.notarstefano@univpm.it (V.N.); e.damiani@univpm.it (E.D.); 2Department of Agricultural, Food and Environmental Sciences-D3A, Università Politecnica delle Marche, Via Brecce Bianche, I-60131 Ancona, Italy; r.foligni@univpm.it

**Keywords:** ATR-FTIR, tea, *Camellia sinensis*, hot and cold infusions, extraction, polyphenols

## Abstract

ATR-FTIR (Attenuated Total Reflectance Fourier Transform InfraRed) spectroscopy, combined with chemometric, represents a rapid and reliable approach to obtain information about the macromolecular composition of food and plant materials. With a single measurement, the chemical fingerprint of the analyzed sample is rapidly obtained. Hence, this technique was used for investigating 13 differently processed tea leaves (green, black and white) all grown and processed in European tea gardens, and their vacuum-dried tea brews, prepared using both hot and cold water, to observe how the components differ from tea leaf to the in-cup infusion. Spectra were collected in the 1800–600 cm^−1^ region and were submitted to Principal Component Analysis (PCA). The comparison of the spectral profiles of leaves and hot and cold infusions of tea from the same country, emphasizes how they differ in relation to the different spectral regions. Differences were also noted among the different countries. Furthermore, the changes observed (e.g., at ~1340 cm^−1^) due to catechin content, confirm the antioxidant properties of these teas. Overall, this experimental approach could be relevant for rapid analysis of various tea types and could pave the way for the industrial discrimination of teas and of their health properties without the need of time-consuming, lab chemical assays.

## 1. Introduction

Tea is one of the simplest and oldest pleasures in the world. Tea is stimulating drink, a symbol of welcome and hospitality and, after water, the most-drunk beverage globally. The main producing countries are China, India, Kenya and Sri Lanka, which together represent over approximately 75% of world production, but, although cultivated primarily in Asia until the 1800s, growing demand made tea production spread across the world, resulting in over 40 producing countries, including Turkey, Vietnam, Indonesia, Japan, Nepal, Korea, Malawi, South Africa, Mozambique, Brazil and Argentina. Each country has its own production areas and each its own unique and unrepeatable terroir with particular soil conditions, altitude, climate, surrounding environment, which produce completely different teas in terms of character, aroma and flavour [1].

Over the last 150 years, there have been several experiments in Europe growing the *Camellia sinensis* plant and making tea from it. Some early experiments can be found in Portugal and France, but lately, different European countries have begun manufacturing tea, leading to the creation of the Association “Tea Grown in Europe” (EuT) in 2016 [2]. With the aim to study the influence that the diverse pedoclimatic (soil type, sun exposure and rainfall) and manufacturing conditions have on the composition and on the nutritional properties of the beverage, we previously studied the antioxidant capacity and the essential and potentially toxic elements content of black, green, yellow, oolong and white teas produced across the European territory from different *Camellia sinensis* cultivars [3,4,5]. The information obtained showed that teas grown in Europe are good quality, with no health hazard, endowed with levels of health-promoting polyphenols, flavonoids and antioxidant capacities similar to those grown in other parts of the world, suggesting that European teas represent a valid alternative to the more common Chinese, Indian and African teas.

Fourier Transform InfraRed (FTIR) spectroscopy is a valid and reliable approach to obtain information about the macromolecular composition of a biological matrix, including food. This analytical technique can provide with a single measurement the complete chemical fingerprint of the analysed sample, with the added advantage of being fast and nondestructive [6]. In recent years, FTIR spectroscopy has been successfully applied for rapid and reliable testing of plant material, also thanks to the advance of chemometric technologies [7,8]. It could be considered a promising technology to be utilized by plant breeders as well as in a variety of industries where rapid screening of numerous samples for antioxidant and/or phenolic content is required [9]. Several studies have reported the ability of near infrared (NIR) spectroscopy which works in the spectral range 13000–4000 cm^−1^, to predict, with the help of multivariate calibration methods, the presence of different bioactive compounds in tea such as catechins, caffeine, free amino acids and theaflavins [10,11,12,13,14,15]; this technique has been also used to discriminate different tea varieties, also in relation with the geographical origins, and to identify tea grades and tea processing degree [10,16,17,18]. Conversely, to the best of our knowledge, only a few studies can be found in the literature regarding the use of FTIR spectroscopy in the MIR region (spectral range 4000–400 cm^−1^) to investigate tea leaves and tea brews and to highlight differences due to processing, cultivation site and extraction methods [4,16,18,19,20,21,22]. One advantage of FT-MIR spectroscopy is its ability to provide with one single acquisition the real chemical fingerprint of the analysed material, since it is very sensitive to the chemical composition of the sample as it contains the fundamental vibrations of almost all the functional groups [23]. With this background, in the present study, dry leaves of different cultivars of black, green and white teas produced across the European territory were analysed by ATR-FTIR spectroscopy, to obtain a rapid and complete characterization of their overall chemical fingerprint and to explore the potential of this analytical approach in discovering specific chemical features correlated to differently processed teas. In addition, to obtain insights on the influence of the water temperature of the infusion on the properties of the brews, considering that also cold-water steeping is increasingly gaining popularity, especially in the summer months, tea leaves were extracted under different conditions and the obtained results were compared amongst each other and with those of the dry leaves.

## 2. Materials and Methods

### 2.1. Tea Samples

Thirteen tea samples subdivided as follows: six green (G), five black (B) and two white (W), were studied coming from seven tea gardens located across Europe (Table 1) whose leaves had been harvested in the 2021 season. Details regarding the temperature range of these gardens, location, growing season, altitude, humidity and average rainfall can be found in the EuT Association leaflet [2]. The tea samples are described in Table 1 where they are identified with an acronym of two letters indicating the country of origin: Italy = I, Netherlands = N, Switzerland = S, Germany = G, Jersey (UK) = J, Portugal = P and Azores (P) = A, followed by the type of tea: green = G; black = B; white = W. Furthermore, for the infusions, a letter indicating the type (C = cold, H = hot) was added at the end as reported in the figures and in Section 3. The tea leaves were processed after plucking the fresh tea shoots (usually two leaves and a bud) from their respective tea gardens, according to the acceptable guidelines of the tea practice industry [24] and are those reported in [3].

### 2.2. Preparation of Tea Brews

The tea brews were prepared as previously described in [3] using 1.0 g of tea leaves and 50 mL of mineral water (ACQUA SANT’ANNA S.p.A., Vinadio, CN, Italy, with a fixed residue at 180 °C of 22 mg/L and total hardness of 0.98 °F). Before preparing the infusions, the dry tea samples were all milled utilizing a hand-mill to obtain a homogeneous fine powder for each type of tea. This is essential for reducing the variability in extraction efficiency that could arise from different leaf sizes. Briefly, hot brews were prepared using boiled water (95–100 °C) and an infusion time of 5 min, whereas cold brews were prepared using water at room temperature (20–25 °C), followed by agitation for 5 s and refrigeration (4–6 °C) for 16 h. Both types of brews were then filtered through filter paper (Whatman Grade 4 qualitative, from Merck KGaA—Darmstadt, Germany), aliquoted and stored at −20 °C until analysed. For each tea sample, the two brewing methods were performed in triplicate on three separate days.

For the ATR-FTIR analysis, 1 mL samples of the tea brews were evaporated in a vacuum concentrator (Heto Hetovac VR-1, Fairbanks, AK, USA) for 6–8 h until complete dryness.

### 2.3. ATR-FTIR Measurements and Data Analysis

ATR-FTIR measurements were performed both on dry tea leaves from different European countries and on the corresponding vacuum-dried hot and cold brews. The analysis was carried out on a Bruker Invenio-R interferometer equipped with a Platinum ATR accessory mounting a diamond crystal and a Deuterated TriGlycine Sulfate (DTGS) detector (Bruker Optics, Ettlingen, Germany).

Dry tea leaves were crushed in a mortar to obtain a homogeneous powder, which was deposited onto the diamond crystal and gently pressed to obtain a good adhesion to the crystal surface. For the hot and cold tea brews, the dried samples were diluted with 20 µL of deionized water and 4 µL of the obtained solutions were deposited onto the diamond crystal and left to air dry for 30 min. ATR-FTIR spectra were collected at room temperature in the 4000–600 cm^−1^ range (128 scans, 4 cm^−1^ spectral resolution). Before each sample acquisition, the spectrum of the background was collected on the clean diamond crystal under the same conditions. Raw spectra were corrected for atmospheric carbon dioxide and water vapor, vector normalized in the whole spectral range and baseline corrected (OPUS 7.5, Bruker Optics, Ettlingen, Germany). Approximately, 8/10 replicates were analysed for each leaf sample and 3/4 for each brew. All those IR spectra with an intensity of the band at 1035 cm^−1^ lower than 0.08 a.u. were discarded.

Preprocessed spectra of hot and cold tea brews and leaves were submitted to Principal Component Analysis (PCA) with no further preprocessing. PCA was performed in the 1800–600 cm^−1^ spectral range. PCA scores plots, and, in some cases, corresponding PC loadings were obtained (Origin PRO 2018 software).

The average spectrum of the leaves obtained from each tea sample and the corresponding average spectra ± standard deviation were submitted to peak fitting procedure in the range of interest 1800–600 cm^−1^ (containing the vibrational modes of proteins and carbohydrates). The number and position of all the underlying bands were evaluated by Second Derivative minima analysis and fixed during fitting procedure with gaussian functions (GRAMS/AI 9.1, Galactic Industries, Inc., Salem, NC, USA). Peaks assignment was performed according to the scientific literature. The integrated areas of all underlying peaks were obtained and used to calculate specific band area ratios.

## 3. Results and Discussion

The IR spectra of leaves collected from black, green and white European teas and of the corresponding cold and hot brews were analysed to obtain the complete spectral fingerprint of each sample. Moreover, IR spectra were submitted to PCA to highlight differences among differently processed teas, as well as to investigate the effects of water temperature on the corresponding brews.

### 3.1. Tea Leaves

The representative IR spectra of black, green and white tea leaves from European crops are shown in Figure 1. Spectra are reported in the most informative region 1800–600 cm^−1^ and the main absorption peaks are listed below: ~1730 cm^−1^ (stretching vibration of the carbonyl moiety, hemicellulose); ~1700 cm^−1^ (CONR stretching vibration, caffeine); ~1632 cm^−1^ (CONH stretching vibration, protein components); ~1517 cm^−1^ (NH bending vibration); ~1456 cm^−1^, ~1365 cm^−1^ and ~1318 cm^−1^ (bending vibrations of CH_2_ moieties); ~1340 cm^−1^ (C-O stretching combined with phenyl ring vibration, catechins); ~1233 cm^−1^ (phenyl ring breath); ~1146 cm^−1^ and ~1035 cm^−1^ (C-OH and C-O-C stretching vibrations). The assignment of peaks was performed according to the literature [7,8,25,26,27,28,29].

The spectral profiles of leaves from black teas (Figure 1a) are very similar to each other, even if slight differences concerning peaks intensities can be noted; interestingly, all samples display two bands at ~1365 cm^−1^ and ~1318 cm^−1^. Conversely, IR spectra from green tea leaves resulted more different, due to both the intensity and the presence in some samples of a third band at ~1340 cm^−1^, besides those always observed at ~1365 cm^−1^ and ~1318 cm^−1^ (Figure 1b): the 1340 cm^−1^ band could be likely attributed to the C-O bond present in catechins, which are found in lower quantity in black teas since they are oxidized to quinoid oligomers [30]. As regards white tea leaves (Figure 1c), only two samples were analysed coming from Azores (AW) and Germany (GW): interestingly, the spectrum of GW in the region 1300–1400 cm^−1^ is more similar to the green tea ones (with three bands at ~1365 cm^−1^, ~1340 cm^−1^ and ~1318 cm^−1^), while AW resembles the spectral profile of black teas with only two bands at ~1365 cm^−1^ and ~1318 cm^−1^.

To better highlight the differences among the differently processed teas and their country of production, IR spectra of tea leaves were submitted to PCA in the 1800–600 cm^−1^ region. In Figure 2a, the PCA scores plot of IR spectra from all samples is reported: only a tiny segregation is observed along the PC1 axis (56.9% of the total variance explained) between black and green teas, mostly attributable to IG, GG and PG. The PCA performed separately on black (Figure 2b) and green (Figure 2c) teas confirmed what was already observed in the analysis of the IR spectra: no segregation was found among the black tea leaves, except for JB that separates along the PC2 axis (11.5%). A better separation was observed among all green leaves (Figure 2c) with IG and PG differentiating from the others along both axes (84.5%). White teas deserve a separate discussion. In fact, AW and GW displayed (Figure 2d) a complete segregation along the PC2 axis (17%) and the analysis of PC2 loadings revealed that the most relevant differences between the two groups of spectra are ascribable to the bands at ~1640 cm^−1^, ~1540 cm^−1^, ~1340 cm^−1^, ~1145 cm^−1^ and ~1035 cm^−1^ (Figure 2d). It is important to note that the band at 1340 cm^−1^ (attributed to catechin) that characterizes the difference between green and black teas, contributes to sample separation.

The following specific band area ratios were statistically analysed: A_1632_/A_1035_ (calculated as ratio between the areas of the bands at 1632 cm^−1^ representative of the CONH moiety in alkaloids, and 1035 cm^−1^, representative of cellulose); A_1146_/A_1035_ (calculated as ratio between the areas of the bands at 1146 cm^−1^ representative of the COH moiety in polyphenols, and 1035 cm^−1^, representative of cellulose). The ratio A_1632_/A_1035_ represents the relative amount of alkaloids (caffeine) with respect to cellulose; in general, in almost all cases, statistically significant higher values were observed in black teas than green ones (*p* < 0.05) (Figure 3a). The ratio A_1146_/A_1035_ represents the relative amount of polyphenols with respect to cellulose; in general, lower values in black teas than green ones are observed (*p* < 0.05), except for G and J, whose differences are not significant (Figure 3b).

### 3.2. Tea Brews

The representative IR spectra of brews are shown in Figure 4. Also in this case, the spectra are reported in the most informative region 1800–600 cm^−1^: the main absorption peaks are the same as those already described for the tea leaves spectra (Section 3.1) with more convoluted bands in the 1300–1500 cm^−1^ region. In cold brews, differences attributable to the country of production are mainly observed in the black teas (Figure 4a), whereas the spectra of the green ones are less distinguishable (Figure 4b). The spectral profiles of the hot brews are more similar to each other with respect to the cold brews: however, also in this case, the spectral differences are more evident in the black teas (Figure 4d) compared to the green ones (Figure 4e). As already reported above for the leaves, the spectra of brews from AW e GW white teas are very different from each other, with AWC being more similar to black teas and GWC to green ones; this is more evident in cold than in hot brews (Figure 4c,f). Noteworthy is the appearance of a band at ~1010 cm^−1^ (Figure 4d) only in hot black brews (for discussion see Section 3.3).

The PCA scores plot calculated on IR spectra of all cold brews is displayed in Figure 5a: a separation between black and green infusions can be observed. In addition, Figure 5b, which reports the PCA score plot of cold brews from black teas, confirms that the brews are quite different from each other with NBC located in the upper left corner of the plot and GBC in the central lower part (PC1: 73.3% and PC2: 13.8% of the total variance explained). A lesser segregation was observed among the green ones (Figure 5c) with only PGC separating well (PC1: 67.1% and PC2: 17.8%). A very good segregation (PC1: 95.4% and PC2: 3.3%) can be observed for AWC and GWC (Figure 5d). For each group of brews, the respective loading plots are also displayed.

In the PCA scores plot of all hot brews (Figure 6a) a good separation between black and green infusions can be observed (PC1: 68.3% and PC2: 17.5% of the total variance explained). Furthermore, the Azorean white brew (AWH) separates well from all the others, whereas the other white tea infusion (GWH) colocalizes with the green ones. A separation can be observed also among black brews when analysed separately (Figure 6b): NBH completely differentiates from all the rest (PC1: 56.7% and PC2: 6.0%). For the green teas (Figure 6c), no separation was found except for PGH that localizes in the right-hand side of the plot (PC1: 42.5% and PC2: 29.7%). Again, the white tea brews appear completely separated along the PC1 axis (Figure 6d) (96.5%).

### 3.3. Comparison between the IR Spectral Featuress of Leaves and Their Corresponding Brews

The comparison of the spectral profiles between leaves and hot and cold infusions of tea coming from the same country, emphasizes how they differ in relation to the different spectral regions. For all the tea gardens the trend observed was similar, and hence for the sake of brevity, only the German case is shown and discussed (Figure 7).

With regards to the black tea GB (Figure 7a), in the region 1000–1100 cm^−1^ typical of carbohydrates vibrations, the dry leaves show a broad peak at ~1040 cm^−1^ that shifts to ~1050 cm^−1^ in the cold extract (GBC); moreover, the spectrum of the hot extract (GBH) exhibits an additional peak at ~1010 cm^−1^. Considering that in this region, the vibrational modes of the C-OH and C-O-C bonds are found, these bands can be assigned both to mono and polysaccharides such as cellulose [31] and to flavonoid O-glycosides [25]. Probably, in the infusions, the decrease in the cellulose content, that cannot be extracted by water, permits the appearance of the bands associated to glycosides. Furthermore, the band at ~1010 cm^−1^ could be attributed to the presence of quinoid oligomers (theaflavins and thearubigins) deriving from catechins oxidation during black tea processing, that are more efficiently extracted by hot rather than cold water and can thus appear only in the hot brew spectrum.

The analysis of the spectra of green tea shows very little differences between leaves, and infusions, except for a slight decrease in the band at ~1040 cm^−1^ in the brews. In the green tea, the absence of the band at ~1010 cm^−1^, both in hot and cold brews, confirms its attribution to quinoid oligomers that are usually not formed during green tea processing.

Finally, concerning the region 1500–1800 cm^−1^ several considerations can be made: (i) for all the types of tea, a band at ~1730 cm^−1^ attributed to hemicellulose [28] appears only in the spectra of the dry leaves, confirming that it is not transferred to the brews independently of the water temperature; (ii) in all black teas, there is a shift of the band at ~1632 cm^−1^ in the leaves to ~1600 cm^−1^ in the infusions; (iii) in all the infusions (for all the tea types) the region between 1505–1560 cm^−1^ is very different from the leaves with only one band appearing around ~1515 cm^−1^ instead of three (~1510 cm^−1^, ~1535 cm^−1^, ~1550 cm^−1^) which are probably hidden by the increase in the convoluted band at ~1600 cm^−1^.

### 3.4. Comparison of IR Spectral Featuress with Antioxidant Activity

The difference in the spectral features of the different samples confirm our previously reported outcomes regarding the nutraceutical properties of these teas [3]: in general, the antioxidant activity and the polyphenolic content measured for these teas are lower in black teas with respect to green ones. This could be likely attributed to the transformation of catechins and other polyphenols to quinoid oligomers during fermentation [32].

These results are confirmed by the following evidences on the tea leaves: (i) in black teas which should be poor in catechins, the band at ~1340 cm^−1^ is lower than in green ones; (ii) German white tea leaves (GW) that display an IR spectrum similar to the green ones (all characterized by the presence of the band at ~1340 cm^−1^) is endowed with greater antioxidant capacity than the Azorean white tea (AW) whose spectrum is similar to that of black teas; (iii) the trend in the band area ratios of PC loadings A_1146_/A_1035_ that represents the relative amount of polyphenols with respect to cellulose (Figure 3b) is similar to those reported in our previous published results [3] and (iv) PG tea that always separates well in all the PCA elaborations (Figure 2c, Figure 5c and Figure 6c) resulted the tea with the greatest antioxidants activity [3].

Furthermore, it is important to note that the IR analysis was able to highlight that the extraction of black tea is significantly influenced by the water temperature: cold water is not able to extract some components which are responsible both for the characteristic taste of black tea but also for its astringency [33].

In this study, the observed differences in catechin content revealed by the IR analysis could aid in understanding the health benefits of the different tea types. Catechins are the major phenolic compounds in tea infusions [34] and drinking tea is linked to body weight control and improvement of several chronic diseases through its antioxidant and anti-inflammatory properties, but also through its beneficial effects on gut microbiota [35,36]. Furthermore, it is assumed that these beneficial effects can be achieved by moderate tea consumption (3–5 cups per day) [37,38]. Since green tea is endowed with a higher content of catechins, it is perceived as a healthier beverage than the more oxidized teas such as black tea [39], although both appear to bestow the same beneficial effects on blood vessel function [40].

## 4. Conclusions

To the best of our knowledge, this is one of the first studies which exploits ATR-FTIR spectroscopy to characterize the chemical fingerprinting of teas on both leaves and their corresponding hot and cold infusions. Although the number of samples was limited, due to the numerosity of European tea gardens that participated in the study, by using this experimental approach, we were, however, able to show that there are some variations not only among the differently processed teas, but also among the European countries in which they are grown and the brewing methods adopted. Noteworthy, from the spectral features, distinct differences between the dry leaves and their infusions were observed as would be expected, and furthermore, that hot tea brews actually differ from cold ones. This experimental approach could be of relevance for rapid analysis of teas coming from different regions of the world and could pave the way for the industrial discrimination of different teas and of their health properties without the need to resort to time-consuming wet-lab biochemical/chemical assays.

## Figures and Tables

**Figure 1 foods-13-00109-f001:**
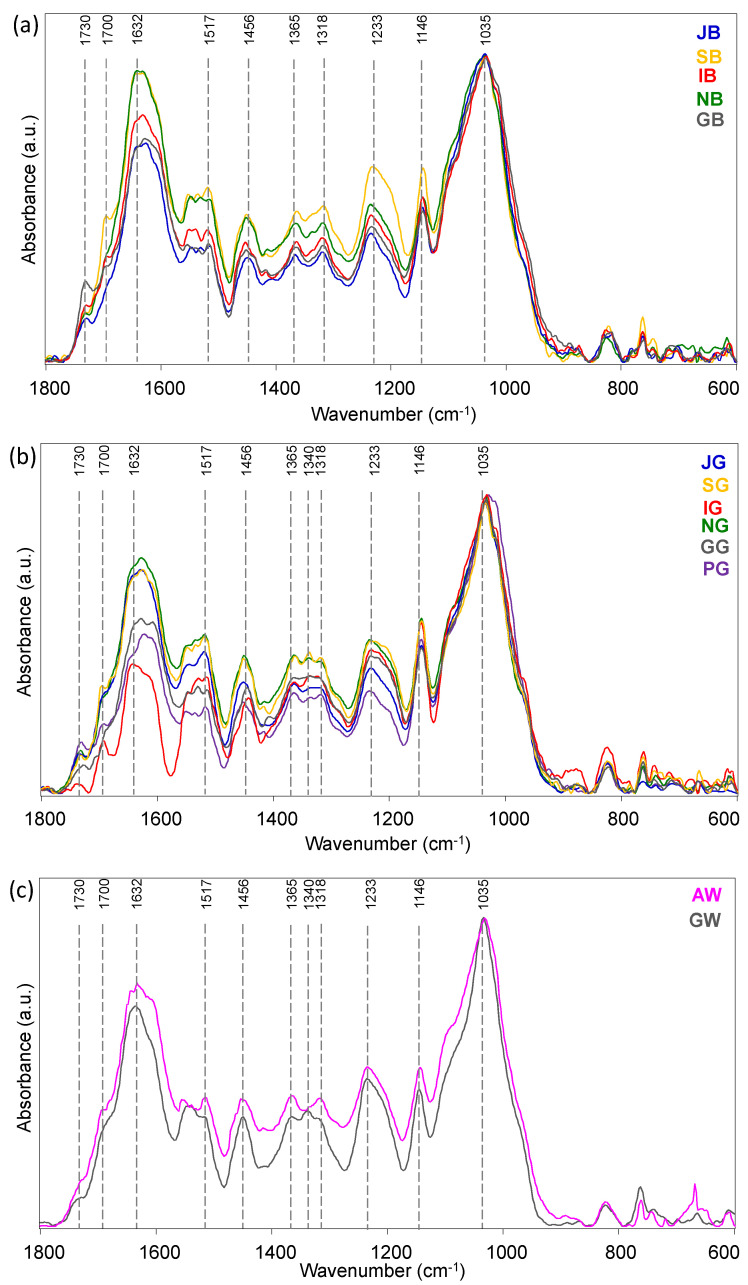
Representative IR spectra of leaves from: (**a**) black, (**b**) green and (**c**) white teas from European crops. IR spectra are reported in the most informative spectral range 1800–600 cm^−1^.

**Figure 2 foods-13-00109-f002:**
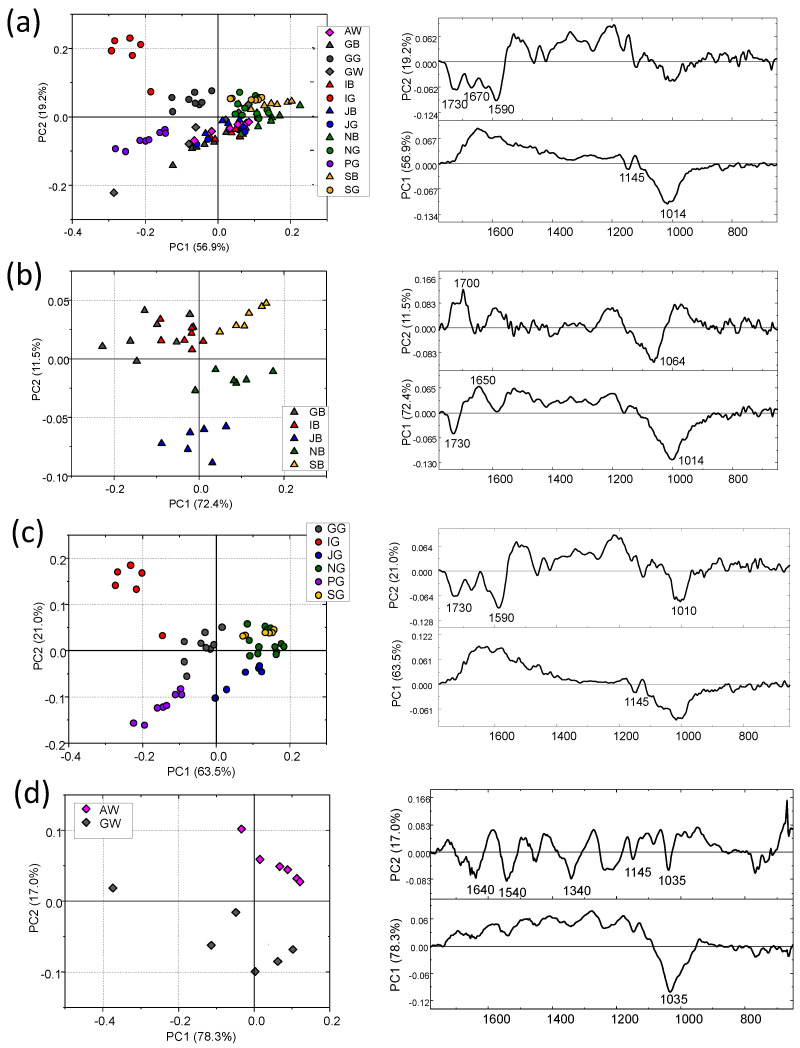
PCA scores plots and corresponding PC1 and PC2 loading plots calculated on the IR spectral populations of dry leaves from: (**a**) all tea samples; (**b**) black teas; (**c**) green teas and (**d**) white teas. PCA was performed in the 1800–600 cm^−1^ range.

**Figure 3 foods-13-00109-f003:**
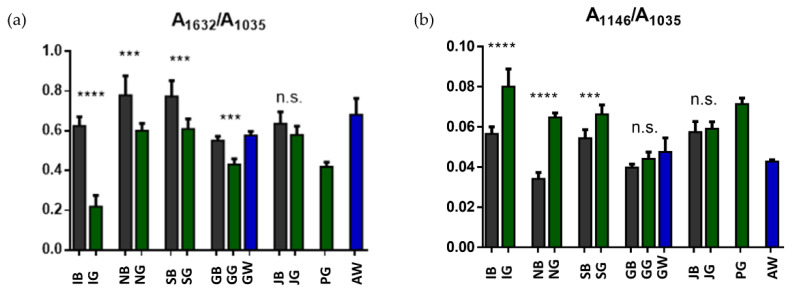
Univariate analysis of meaningful band area ratios: (**a**) A_1632_/A_1035_ (calculated as ratio between the areas of the bands at 1632 cm^−1^ representative of the CONH moiety in alkaloids, and 1035 cm^−1^, representative of cellulose); (**b**) A_1146_/A_1035_ (calculated as ratio between the areas of the bands at 1146 cm^−1^ representative of the COH moiety in polyphenols, and 1035 cm^−1^, representative of cellulose). Statistical analysis was performed with the software package Prism 6.0 (GraphPad Software, Inc., San Diego, CA, USA). All data were presented as mean ± standard deviation (S.D.). Statistical significance among groups was evaluated using Student’s *t*-test. Statistical significance was set at *p* < 0.05 (***, *p* < 0.001; ****, *p* < 0.0001, n.s., not significant).

**Figure 4 foods-13-00109-f004:**
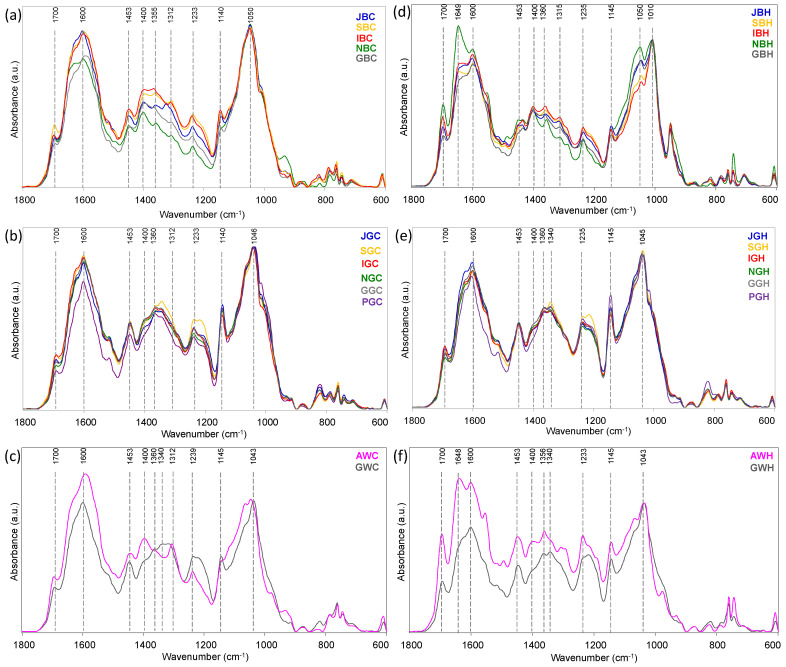
Representative IR spectra of cold ((**a**): black, (**b**): green, (**c**): white) and hot brews ((**d**): black, (**e**): green, (**f**): white) of teas from European crops. IR spectra are reported in the most informative spectral range 1800–600 cm^−1^.

**Figure 5 foods-13-00109-f005:**
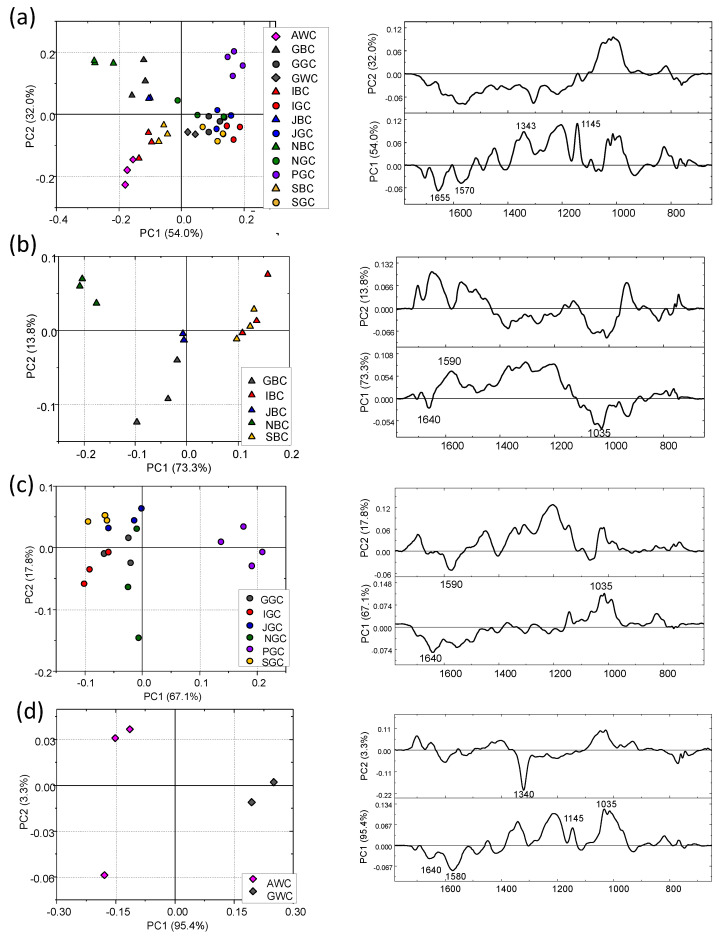
PCA scores plots and corresponding PC1 and PC2 loading plots calculated on the IR spectral populations of cold tea brews from: (**a**) all tea samples; (**b**) black teas; (**c**) green teas and (**d**) white teas. PCA was performed in the 1800–600 cm^−1^ range.

**Figure 6 foods-13-00109-f006:**
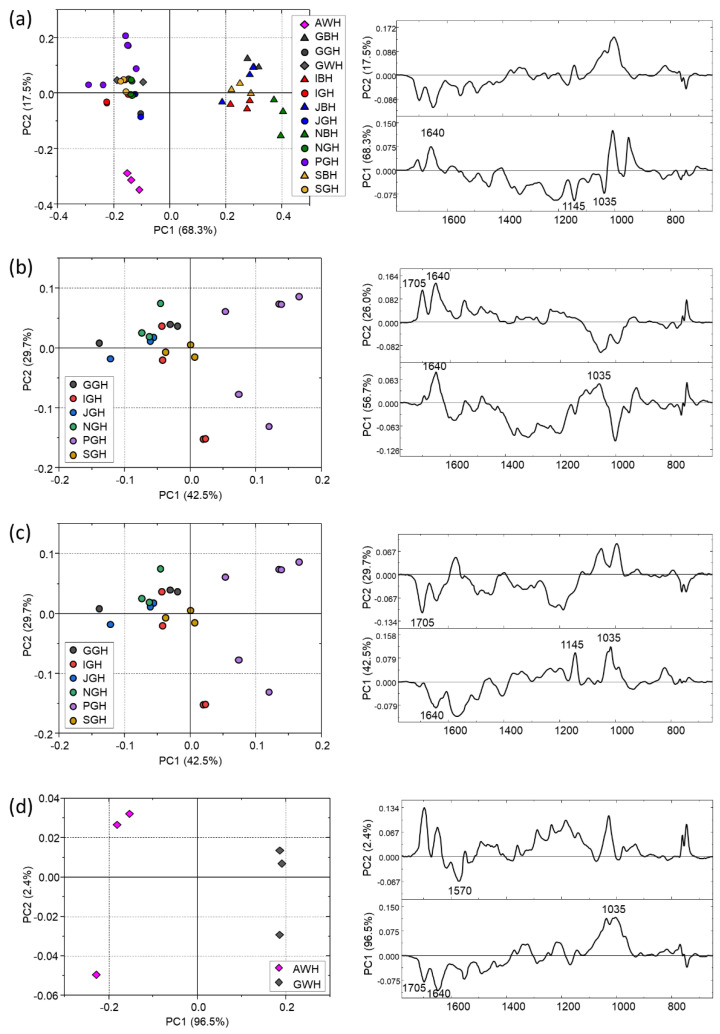
PCA scores plots and corresponding PC1 and PC2 loading plots calculated on the IR spectral populations of hot tea brews from: (**a**) all tea samples; (**b**) black teas; (**c**) green teas and (**d**) white teas. PCA was performed in the 1800–600 cm^−1^ range.

**Figure 7 foods-13-00109-f007:**
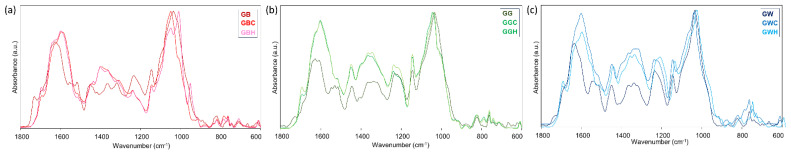
Representative IR spectra of leaves, cold and black brews ((**a**): black, (**b**): green, (**c**): white) of teas from Germany. IR spectra are reported in the most informative spectral range 1800–600 cm^−1^.

**Table 1 foods-13-00109-t001:** List of the analysed teas, reporting the tea garden name, home country, tea type and labelling of the European teas.

European Garden	Country	Type	Label
Jersey Fine Tea	United Kingdom	Black	JB
		Green	JG
Casa del Tè Monte Verità	Switzerland	Black	SB
		Green	SG
Compagnia del Lago Maggiore	Italy	Black	IB
		Green	IG
Het Zuyderbald	Netherlands	Black	NB
		Green	NG
Tschanara Tea Garden	Germany	Black	GB
		Green	GG
		White	GW
Chà Camelia	Portugal	Green	PG
Agrarian Devt. Services Sao Miguel	Azores (Portugal)	White	AW

## Data Availability

Data is contained within the article.

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
