# Peer review of "First ATR-FTIR Characterization of Black, Green and White Teas (Camellia sinensis) from European Tea Gardens: A PCA Analysis to Differentiate Leaves from the In-Cup Infusion"

_foods, 2023, doi:10.3390/foods13010109_

Round 1
Reviewer 1 Report
Comments and Suggestions for Authors
The authors investigated the dissolved matter characteristics of tea in hot and cold brewing modes. Readers may be more interested in learning about the differences between these two modes of tea brewing, such as composition and dissolution rate. In addition, there is less useful information for readers from this manuscript. Could the authors add attribution of features to the infrared spectra of tea leaves and tea brews and then analyze the differences or characteristics of the different teas from a fingerprint spectroscopy perspective?
Other details are commented below:
Lines 59-62: Many researchers have demonstrated that near-infrared spectroscopy can be used to analyze the chemical composition of tea and tea products, as well as to identify the origin and variety of tea. Authors may add comments.
Lines 85-86: Room temperature water is 4-6°C?
Author Response
The authors investigated the dissolved matter characteristics of tea in hot and cold brewing modes. Readers may be more interested in learning about the differences between these two modes of tea brewing, such as composition and dissolution rate.
We added the complete description of the two modes of brewing in the Material and Methods section (2.2 Preparation of tea brews) and we added an explanation of the interest of these different mode of extraction in the Introduction (Lines 85-88)
In addition, there is less useful information for readers from this manuscript. Could the authors add attribution of features to the infrared spectra of tea leaves and tea brews and then analyze the differences or characteristics of the different teas from a fingerprint spectroscopy perspective?
We believe there has been a misunderstanding since the most relevant IR peaks found in the spectra of tea leaves and brews were already reported in the Results and Discussion section 3.1 (see lines 168-174) together with the vibrational modes and the corresponding biological meaning. The observed spectral features were then analyzed to highlight the differences in the composition of leaves and brews (Section 3.3. Comparison between the IR spectral featuress of leaves and their corresponding brews).
Other details are commented below:
Lines 59-62: Many researchers have demonstrated that near-infrared spectroscopy can be used to analyze the chemical composition of tea and tea products, as well as to identify the origin and variety of tea. Authors may add comments. Cozzolino 2022, Nagy 2022
We agree with the reviewer, and lines 62-78 have been modified to highlight the presence of several studies that have used the NIR spectroscopy for the analysis of tea. Nevertheless, we would like to point out that NIR and MIR spectroscopies are quite different in terms of results. We are conscious that NIR spectroscopy is used to study tea samples. Nevertheless, we are certain that the collection of spectra in the MIR region can give a more precise and detailed characterization of samples. In fact, in the NIR region, we can mostly find bands attributable to vibrations of bonds with an H atom; moreover, the NIR bands are often broad and hence NIR spectra can result indistinguishable from each other. Conversely, the MIR region is very sensitive to the chemical composition of the sample, since it contains the fundamental vibrations of almost all the functional groups, and hence it can provide the real chemical fingerprint of the analyzed material.
Lines 62-80 “Several studies have reported the ability of NIR spectroscopy which works in the spectral range 13000-4000 cm-1, to predict, with the help of multivariate calibration methods, the presence of different bioactive compounds in tea such as catechins, caffeine, free amino acids, theaflavins {Chen, 2009;Chen, 2008;Cozzolino, 2022;Wang, 2018;Wen, 2023;Zareef, 2018}; this technique has been also used to discriminate different tea varieties, also in relation with the geographical origins, and to identify tea grades and tea processing degree {Chen, 2009;Esteki, 2022;Song, 2021;Zhou, 2020}. Conversely, to the best of our knowledge, only few studies can be found in the literature regarding the use of FTIR spectroscopy in the MIR region (spectral range 4000-600 cm-1) to investigate tea leaves and tea brews and to highlight differences due to processing, cultivation site and extraction methods {Aboulwafa, 2019;Arifah, 2022;Budı́nová, 1998;Carloni, 2023;Carvalho, 2021;Esteki, 2022;Zhou, 2020}. One advantage of FT-MIR spectroscopy is its ability to provide with one single acquisition the real chemical fingerprint of the analyzed material, since it is very sensitive to the chemical composition of the sample as it contains the fundamental vibrations of almost all the functional groups {Stuart, 2004}.
Lines 85-86: Room temperature water is 4-6°C?
We apologize for this incorrect explanation of the brewing temperature used. Water was added to the tea leaves at room temperature and the leaves were left to infuse for 16 h at 4-6°C. The description of the procedure has been added in the Material and Methods section (2.2 Preparation of tea brews).

Reviewer 2 Report
Comments and Suggestions for Authors
The paper "First ATR-FTIR characterization of black, green and white teas (Camellia sinensis) from European tea gardens: how the components differ from leaf to the in-cup infusion" by E. Giorgini et al. provides a potentially interesting study that, however, presents several weaknesses.
The title can be misleading: PCA is not mentioned in the title, and "... how the components differ ..." seems more related to chemical composition than to spectral differences, and this should be corrected.
Abstract. In the abstract, it appears that dissolutions are measured directly. It should be indicated that it is the dissolved solid phase after removal of water.
Line 72 and the following lines. 2.1. Tea samples and preparation of infusions.
Thirteen samples from three different groups is not a significant number to obtain conclusions.
Regarding hot and cold infusions, were they made with which type of water? In reference 3, mineral water was indicated, but there are many types (for example, hard or soft water?).
The same for the post-treatment of the infusion, was it filtered as in reference 3?
Line 93 and following. 2.2. ATR-FTIR measurements and data analysis.
Were the pulverized samples sieved in any way to ensure a minimum particle size?
Lines 101-... Regarding the evaporated liquid samples, would it not be better to use a polar solvent, for example, a type of alcohol, to avoid the 30-minute bottleneck when using water?
Line 109: by what criteria was a spectrum considered to be an outlier? Indicate methodology or reference, since the elimination of an outlier spectrum is not a trivial exercise.
Line 115: I understand that the average spectrum is used. What is not so clear is why the mean spectrum was not kept for the rest of the study. The differences in homogeneous samples are due to the technique used, and its high sensitivity due to the small amount of sample measured, not to the difference in samples (if they were at least samples from the same location taken at different times or degrees of maturation it would make sense). This introduces an artificial source of variability that distorts the PCA study.
Figure 2d) Two-sample PCA does not make sense.
Lines 256-260. The description is correct, but there is no explanation of the chemical reasons for these changes.
There is no record of the loadings of the complete data set, just one cold and one hot different group infusions.
Reference [3] appears to contain chemical composition data for these samples. It would be of much more interest to relate the spectral information to the chemistry with FTIR, potentially more interesting than the UV-visible used there.
Therefore, these data require an increase in the number of samples and a thorough revision to provide something of interest to this journal.
Author Response
The paper "First ATR-FTIR characterization of black, green and white teas (Camellia sinensis) from European tea gardens: how the components differ from leaf to the in-cup infusion" by E. Giorgini et al. provides a potentially interesting study that, however, presents several weaknesses.
The title can be misleading: PCA is not mentioned in the title, and "... how the components differ ..." seems more related to chemical composition than to spectral differences, and this should be corrected.
The differences in the spectral profiles of tea groups obviously derive from a different chemical composition. However, the title has been changed. “First ATR-FTIR characterization of black, green and white teas (Camellia sinensis) from European tea gardens: a PCA analysis to differentiate leaves from the in-cup infusion.”
Abstract. In the abstract, it appears that dissolutions are measured directly. It should be indicated that it is the dissolved solid phase after removal of water.
It has now been specified that tea brews were evaporated (dried out) before the analysis (line 17).
Line 72 and the following lines.
2.1. Tea samples and preparation of infusions.
Thirteen samples from three different groups is not a significant number to obtain conclusions.
We agree with the referee that an increase in the number of samples with a thorough revision would be of additional value. However, not all the European tea growers of the EuTA that we contacted decided to collaborate with us: only 7 gardens sent us their samples. Furthermore, it was not possible to have more than 3 different groups, since European tea gardens are only concentrated on producing green and black teas and only a couple also produce white teas. However, despite this limited number, both in terms of samples and tea groups, we decided to go ahead with our project, and indeed, this preliminary study was worthwhile since the results so far obtained indicate that ATR-FTIR spectroscopy is appropriate and informative and could pave the way to being a new method for discriminating the different teas based on the processing methods, brewing method, and location, of the different European tea gardens, which has never been reported previously. A comment regarding this issue has been added in the Conclusions section.
Lines 349-351: “Although the number of samples was limited, due to the numerosity of European tea gardens that participated in the study, by using this experimental approach, we were however able to show that there are some variations, not only among the differently processed teas, but also among the European countries in which they are grown, and the brewing methods adopted.”
Regarding hot and cold infusions, were they made with which type of water? In reference 3, mineral water was indicated, but there are many types (for example, hard or soft water?).
We added the complete description of the two modes of brewing in the Material and Methods section (2.2 Preparation of tea brews) with the specification of the type of Mineral water.
Lines 112-113: “(ACQUA SANT’ANNA S.p.A., Vinadio, CN, Italy, with a fixed residue at 180 °C of 22 mg/L and total hardness of 0.98 °f)”.
The same for the post-treatment of the infusion, was it filtered as in reference 3?
We added the complete description of the two modes of brewing in the Material and Methods section (2.2 Preparation of tea brews).
Line 93 and following.
2.2. ATR-FTIR measurements and data analysis.
Were the pulverized samples sieved in any way to ensure a minimum particle size?
Because of the nature of the dried tea leaf samples, they can be easily pulverized and the powder obtained is homogenous, therefore it was not necessary to sieve the sample to obtain a minimum particle size. Sieving is mandatory if the sample is hard, such as bones, teeth, etc. because during pulverization it is difficult to obtain a homogenous powder. Moreover, the high signal-to-noise ratio displayed by almost all spectra, confirms that the tea powder had completely adhered to the ATR diamond surface.
Lines 101-... Regarding the evaporated liquid samples, would it not be better to use a polar solvent, for example, a type of alcohol, to avoid the 30-minute bottleneck when using water?
We would like to point out that the tea brews were water solutions, and hence we preferred to use water instead of an organic solvent for resolubilizing them. Moreover, when using water, the drying process can be easily followed by checking the disappearance of the combination band of water detectable at ca. 2130 cm-1, a spectral region free from other absorptions.
Line 109: by what criteria was a spectrum considered to be an outlier? Indicate methodology or reference, since the elimination of an outlier spectrum is not a trivial exercise.
We agree with the reviewer, the term was misleading, hence a more detailed procedure for outlier exclusion has been explained in the text.
Lines 144-145: “All those IR spectra with an intensity of the band at 1035 cm-1 lower than 0.08 a.u. were discarded.”
Line 115: I understand that the average spectrum is used. What is not so clear is why the mean spectrum was not kept for the rest of the study. The differences in homogeneous samples are due to the technique used, and its high sensitivity due to the small amount of sample measured, not to the difference in samples (if they were at least samples from the same location taken at different times or degrees of maturation it would make sense). This introduces an artificial source of variability that distorts the PCA study.
We understand the point of the reviewer. We would not have used PCA if the replicates were obtained by repeated measurements, since we are aware that other statistical procedures would be necessary in that case. In our study, the replicates are referred to samples (replicates of leaves or extracts), hence we decided to include them. In our opinion, including replicates better represents the variance in the dataset; moreover, the obtained loadings, which are crucial in PCA applied to spectroscopy data, are weighted, among all, by the number of spectra in each identified cluster, hence the decrease in the number of elements within the clusters may lead to misinterpretations.
Figure 2d) Two-sample PCA does not make sense.
We do not agree with this statement. PCA applied to spectroscopy not only provides spectral groupings with similar variability, but also and especially it provides loadings, which offer the most discriminant spectral features. In this light, as widely reported, like for example in Bonnier 2012 (https://doi.org/10.1039/C1AN15821J), to obtain a clear loading displaying the spectral origin of differences between two groups, the best results are provided by pair-wise analyses.
Lines 256-260. The description is correct, but there is no explanation of the chemical reasons for these changes.
IR spectra are often composed of convoluted bands due to the overlapping of different peaks, close to each other as regards the wavenumber, but attributable to precise and different vibrational modes. Moreover, the position of IR peaks can be influenced by the chemical environment and by the possible formation of H-bonds. This is also the same in the spectra of tea samples. Hence, it is almost impossible to exactly assign each band and each difference among IR spectra. The spectral interval 1600-700 cm-1 is called fingerprint region because it is characteristic of the studied matrix as a whole, and hence it is not possible to explain which compound determines each single absorption.
There is no record of the loadings of the complete data set, just one cold and one hot different group infusions.
We agree with the reviewer, and we have accordingly modified Figures 2, 5 and 6 adding all the PC1 and PC2 loading plots. However as already stated above, in PCA applied to IR spectra, loadings are very informative only when pair-wise PCA is performed, since deviations from the baseline can be directly associated with precise differences in the spectral profiles of the two groups.
Reference [3] appears to contain chemical composition data for these samples. It would be of much more interest to relate the spectral information to the chemistry with FTIR, potentially more interesting than the UV-visible used there.
The study reported in ref [3: Carloni, 2023] was based on the analysis of cold and hot tea infusions and during this study it was not possible at that time to analyze the samples by FT-IR spectroscopy but only by UV-Vis spectroscopy, which however, has been used in several other studies for the characterization and differentiation of tea (https://doi.org/10.1016/j.saa.2012.10.052, https://doi.org/10.1016/j.foodchem.2015.07.022, https://doi.org/10.1051/e3sconf/202126505013). In our follow-up study [4: Carloni, 2023; https://doi.org/10.3390/antiox12111943] regarding differently processed teas from the same tea garden, we were however able to analyze the FTIR features of tea leaves to determine whether the processing techniques could influence the absorbances. Only subsequently, in the present study, we decided to analyze also the spectroscopic characteristics of the tea infusions to understand how these could be influenced by the extraction methods.
Therefore, these data require an increase in the number of samples and a thorough revision to provide something of interest to this journal.
We agree with the referee that an increase in the number of samples with a thorough revision would be of additional value. However, not all the European tea growers of EuTA that we contacted decided to collaborate with us: only 7 gardens sent us their samples. Furthermore, it was not possible to have more than 3 different groups, since European tea gardens are only concentrated on producing green and black teas and only a couple also produce white teas. However, despite this limited number, both in terms of samples and tea groups, we decided to go ahead with our project, and indeed, this preliminary study was worthwhile since the results so far obtained indicate that ATR-FTIR spectroscopy is appropriate and informative and could pave the way to being a new method for discriminating the different teas based on the processing methods, brewing method, and location, of the different European tea gardens, which has never been reported previously. A comment regarding this issue has been added in the Conclusions section.
Lines 349-351: “Although the number of samples was limited, due to the numerosity of European tea gardens that participated in the study, by using this experimental approach, we were however able to show that there are some variations, not only among the differently processed teas, but also among the European countries in which they are grown and the brewing methods adopted.”

Reviewer 3 Report
Comments and Suggestions for Authors
This work presents a study using ATR-FTIR spectroscopy combined with chemometrics to analyze black, green, and white tea leaves and their brews from European tea gardens. The aim was to observe how the components of these teas differ from the leaf to the in-cup infusion using both hot and cold water. The study investigated 13 differently processed tea leaves and their brews, collecting spectra in the 1800-600 cm‒1 region and submitting them to PCA. This analysis highlighted the differences between leaves and infusions, as well as among teas from different countries. The study found that the spectral profiles of leaves and hot and cold infusions of tea from the same country emphasize how they differ in relation to different spectral regions. The changes observed due to catechin content confirmed the antioxidant properties of these teas. The study concludes that this experimental approach could be relevant for rapid analysis of various tea types and could pave the way for the industrial discrimination of teas and their health properties without the need for time-consuming lab chemical assays.
1.Expand on the statistical analysis section, particularly regarding the PCA. Include more details on the variance explained by each principal component and how they contribute to differentiating the tea samples.
2.Include additional graphical representations, such as scatter plots or heat maps, to visually demonstrate the PCA results and the spectral differences among tea types.
3.Clarify the temperature control methods used during the infusion process. Temperature can significantly impact the extraction of compounds, which may affect the results.
4.Elaborate on the study's implications for understanding the health benefits of different tea types, especially in light of the observed differences in catechin content.
5.Suggest a brief discussion on how the spectral profiles of European teas might compare with those from traditional tea-growing regions in Asia.
Author Response
This work presents a study using ATR-FTIR spectroscopy combined with chemometrics to analyze black, green, and white tea leaves and their brews from European tea gardens. The aim was to observe how the components of these teas differ from the leaf to the in-cup infusion using both hot and cold water. The study investigated 13 differently processed tea leaves and their brews, collecting spectra in the 1800-600 cm‒1 region and submitting them to PCA. This analysis highlighted the differences between leaves and infusions, as well as among teas from different countries. The study found that the spectral profiles of leaves and hot and cold infusions of tea from the same country emphasize how they differ in relation to different spectral regions. The changes observed due to catechin content confirmed the antioxidant properties of these teas. The study concludes that this experimental approach could be relevant for rapid analysis of various tea types and could pave the way for the industrial discrimination of teas and their health properties without the need for time-consuming lab chemical assays.
1.Expand on the statistical analysis section, particularly regarding the PCA. Include more details on the variance explained by each principal component and how they contribute to differentiating the tea samples.
We agree with the reviewer and we have added the variance explained along with the text.
2.Include additional graphical representations, such as scatter plots or heat maps, to visually demonstrate the PCA results and the spectral differences among tea types.
We have modified Figures 2, 5 and 6 adding to the corresponding PCA score plots (scatter plots) the PC1 and PC2 loading plots.
3.Clarify the temperature control methods used during the infusion process. Temperature can significantly impact the extraction of compounds, which may affect the results.
We added the complete description of the two modes of brewing in the Material and Methods section (2.2 Preparation of tea brews).
4.Elaborate on the study's implications for understanding the health benefits of different tea types, especially in light of the observed differences in catechin content.
This aspect has been elaborated in section 3.4, last paragraph.
“In this study, the observed differences in catechin content revealed by the IR analysis could aid in understanding the health benefits of the different tea types. Catechins are the major phenolic compounds in tea infusions (Zhao et al., 2019) and drinking tea is linked to body weight control and improvement of several chronic diseases through its antioxidant and anti-inflammatory properties, but also through its beneficial effects on gut microbiota (Abiri 2023, Saptadip 2020). Furthermore, it is assumed that these beneficial effects can be achieved by moderate tea consumption (3-5 cups per day) (Cheng, 2019; Pérez-Burillo et al., 2018). Since green tea is endowed with a higher content of catechins, it is perceived as a healthier beverage than the more oxidized teas such as black tea (Yang 2022), although both appear to bestow the same beneficial effects on blood vessel function (Fuchs 2014).”
5.Suggest a brief discussion on how the spectral profiles of European teas might compare with those from traditional tea-growing regions in Asia.
The only few studies using FTIR spectroscopy on tea are poorly described and lack in in-depth detail, hence it is difficult to make any comparisons. The only comparison that could be made is with the work by Esteki et al. who analyzed tea samples from Iran. Unfortunately, the comparison between their spectral data and ours is very difficult since the spectra reported in the above cited manuscript are in transmission mode and display a very bad spectral resolution.

Round 2
Reviewer 2 Report
Comments and Suggestions for Authors
The article contains improvements over the first version that have increased its interest.
I congratulate the authors for their efforts.